# Calceolarioside A, a Phenylpropanoid Glycoside from *Calceolaria* spp., Displays Antinociceptive and Anti-Inflammatory Properties

**DOI:** 10.3390/molecules27072183

**Published:** 2022-03-28

**Authors:** Stefano Pieretti, Anella Saviano, Adriano Mollica, Azzurra Stefanucci, Anna Maria Aloisi, Marcello Nicoletti

**Affiliations:** 1National Centre for Drug Research and Evaluation, Istituto Superiore di Sanità, 00161 Rome, Italy; 2ImmunoPharmaLab, Department of Pharmacy, School of Medicine and Surgery, University of Naples Federico II, 80131 Naples, Italy; anella.saviano@unina.it; 3Department of Pharmacy, University “G. d’Annunzio” of Chieti-Pescara, 66100 Chieti, Italy; a.stefanucci@unich.it (A.S.); adriano.mollica@unich.it (A.M.); 4Department Medicine, Surgery and Neuroscience, University of Siena, 53100 Siena, Italy; annamaria.aloisi@unisi.it; 5Department of Environmental Biology, Sapienza University of Rome, 00185 Rome, Italy; marcello.nicoletti@uniroma1.it

**Keywords:** calceolarioside A, nociception, inflammation, phenylpropanoid glycosides, Calceolaria

## Abstract

Phenylpropanoid glycosides are a class of natural substances of plant origin with interesting biological activities and pharmacological properties. This study reports the antinociceptive and anti-inflammatory effects of calceolarioside A, a phenylpropanoid glycoside previously isolated from various Calceolaria species. In models of acute nociception induced by thermal stimuli, such as the hot plate and tail flick test, calceolarioside administered at doses of 1, 5, and 10 μg in the left cerebral ventricles did not modify the behavioral response of mice. In an inflammatory based persistent pain model as the formalin test, calceolarioside A at the high dose tested (100 μg/paw) reduced the licking activity induced by formalin by 35% in the first phase and by 75% in the second phase of the test. In carrageenan-induced thermal hyperalgesia, calceolarioside A (50 and 100 μg/paw) was able to significantly reverse thermal hyperalgesia induced by carrageenan. The anti-inflammatory activity of calceolarioside A was then assessed using the zymosan-induced paw edema model. Calceolarioside A (50 and 100 μg/paw) induced a significant reduction in the edema from 1 to 4 h after zymosan administration. Measuring IL-6, TNFα, and IL-1β pro-inflammatory cytokines released from LPS-stimulated THP-1 cells, calceolarioside A in a concentration-dependent manner reduced the release of these cytokines from THP-1 cells. Taken together, our results highlight, for the first time, the potential and selective anti-inflammatory properties of this natural-derived compound, prompting its rationale use for further investigations.

## 1. Introduction

Plant-derived molecules are a primary source of drugs, also evaluated as possible drugs for anti-inflammatory effects. Research on new plant-derived compounds is driven by the fact that, although several agents are available to treat various inflammatory diseases, their prolonged use can cause serious adverse effects. Nonsteroidal anti-inflammatory drugs (NSAIDs) inhibit the early stages of prostaglandin biosynthesis through cyclooxygenase (COX) inhibition. NSAIDs are essential drugs used to fight inflammation, but chronic use of NSAIDs is linked to cardiovascular, gastrointestinal, and renal toxicities [1,2]. Similarly, the use of corticosteroids leads to hypertension, hyperglycemia, osteoporosis, and stunting [3], and the development of safer anti-inflammatory agents remains a topic of great interest. Many studies on plant species in folk medicine for inflammation have recognized the potential of natural products as possible anti-inflammatory drugs [4]. The anti-inflammatory effects of plant-derived molecules are exerted through their action on critical regulatory molecules, including cyclooxygenase (COX), lipoxygenase (LOX), and cytokines [5,6].

Phenylpropanoid glycosides (PPGs) are acylated glycoconjugates carrying a substituted arylalkyl aglycon, and acylation occurs mainly on the primary hydroxyl group with a residue derived from cinnamoyl. Phenylpropanoid glycosides are secondary metabolites widely distributed in plants with potential therapeutic properties including anti-inflammatory, analgesic, immunomodulatory, and radical scavenging [7,8].

In a previous paper, the common presence of phenylpropanoid glycosides has been reported, i.e., verbascoside (actoside), forsythoside A, isoarenarioside, and calceolarioside A–E from the methanolic extract of several Chilean Calceolaria species [9,10]. The antinociceptive and anti-inflammatory properties of verbascoside and forsythoside A have already been described [7,8], whereas no literature data are available on the effects induced by isoarenarioside or calceolarioside A-E on nociception or inflammation. Therefore, as a starting point in the pharmacological characterization of these compounds, the present study aimed to investigate, for the first time, the antinociceptive potential of calceolarioside A (Figure 1) in thermal and inflammatory models of nociception. We also evaluated calceolarioside A anti-inflammatory effects in an animal model of edema and on LPS-induced cytokine release from human macrophages. The results illustrate the antinociceptive and anti-inflammatory properties of the natural compound calceolarioside A and may help in the development of a new therapeutic strategy for pain and inflammation, particularly in view of its reduction in cytokine release from human macrophages.

## 2. Results

### 2.1. Effects of Calceolarioside A in Animal Models of Nociception

Hot plate and tail-flick tests were employed to assess drug effects on nociception [11]. In our experiments, we administered calceolarioside A at doses of 1, 5, and 10 μg in the left cerebral ventricles of mice who underwent the hot plate and the tail-flick tests. The results obtained in these experiments are reported in Figure 2. After central administration, regardless of the dose used, calceolarioside A did not modify the behavioral response to thermal nociceptive stimuli, both in the hot plate and in the tail-flick test (Figure 2).

Then, we investigated the effects of calceolarioside A in another nociception model, e.g., the formalin test [12]. This test can be considered an inflammatory-based persistent pain model. In this case, we administered calceolarioside A at doses of 10, 50, and 100 μg s.c. in the hind paw of the mouse 30 min before formalin. The results of these experiments are shown in Figure 3.

In the formalin test, administration of calceolarioside A at a dose of 10 μg did not change the animals’ behavioral response induced by formalin. Calceolarioside A (50 μg) induced a nonsignificant reduction (by 14%, compared to the control group) in the licking time induced by the aldehyde in the early phase of the test, while it was able to reduce significantly (by 49%, compared to the control group) the noxious response in the late phase. These effects on the early and late phase induced by calceolarioside A were even more evident after the administration of the 100 μg dose, capable of significantly reducing the nociceptive effect of formalin both in the early (by 31%, compared to the control group) and in the late phase (by 75%, compared to the control group) of the test.

From the results obtained in the formalin test, an effect of calceolarioside A in this model of persistent inflammation is highlighted. However, studies on inflammatory pain use compounds with strong antigenic potential such as carrageenans, sulfated polysaccharides extracted from seaweed. Paw injection of this compound induces thermal and mechanical allodynia and hyperalgesia for at least several hours [11]. Therefore, we used carrageenan to reduce the nociceptive threshold to study the possible anti-hyperalgesic effects of calceolarioside A. Carrageenan induced a strong hyperalgesic effect under our experimental condition, 3–4 h after its administration. Thus, we administered s.c. calceolarioside A at doses of 10, 50, and 100 μg s.c. into the mice hind paw, 2.5 h after carrageenan. The anti-hyperalgesic effects of calceolarioside A could be observed in carrageenan-induced hyperalgesia (Figure 4). Treatment with calceolarioside A at 50 and 100 μg was able to significantly reduce thermal hyperalgesia induced by carrageenan, 3 and 4 h after its administration, compared to the control group (Figure 4).

### 2.2. Effects of Calceolarioside A in Animal Model of Edema

Following the observed antinociceptive effects of calceolarioside A in inflammatory pain models, we aimed to understand if calceolarioside A is able to counteract the formation of edema. Zymosan-induced paw edema is a well-known experimental model to study acute inflammation and the mechanisms of inflammatory pain [13]. Herein, we addressed whether calceolarioside A inhibits zymosan-induced paw edema. Mice were pretreated with different doses of calceolarioside (10, 50, and 100 μg) s.c. into the mice hind paw, 30 min before zymosan s.c. injection in the same manner, and edema were evaluated from 1 to 24 h after the stimulus. As shown in Figure 5, the edema formation was significantly inhibited by calceolarioside A tested at 50 and 100 μg/paw.

### 2.3. Calceolarioside A Effects on LPS-Induced Cytokine Release from Human Macrophage

LPS is a TLR4-specific agonist described as a potent inducer of inflammatory responses in macrophages [14]. To see whether cytokine production is influenced by calceolarioside A, THP-1 cells were treated with an increasing amount of calceolarioside A in the presence of LPS and cytokine secretion was measured from the medium of the cells using ELISA methods. In addition, the toxicity of calceolarioside A on THP-1 cells was accessed to determine the adequate calceolarioside working concentrations. When the cell viability of THP-1 cells was higher than 90%, samples were considered noncytotoxic and adequate for further analysis. The results obtained in these experiments are reported in Figure 6. In agreement with previous reports [14], we found that LPS treatment significantly induced the production of IL-6, TNFα, and IL-1β pro-inflammatory cytokines (Figure 6). While calceolarioside A, at the tested concentrations, did not affect the production of the aforementioned cytokines in THP-1 cells (data not shown) without any effect on macrophages, it significantly reduced the LPS-induced secretion of these cytokines in THP-1 cells in a concentration-dependent manner (Figure 6).

## 3. Discussion

The results of our study demonstrate, for the first time, the selective and stimulus-dependent antinociceptive and anti-inflammatory effects of calceolarioside A. These effects confirm that PPGs are a class of natural substances capable of inducing interesting analgesic and anti-inflammatory effects, observed for various compounds isolated from different plant species [7,8]. As far as calceolarioside A is concerned, very little data are available in the literature on its biological effects. In a previous study, it was found that calceolarioside A induced a dose-related aggregant effect on rabbit platelets, partly dependent upon a calcium-dependent mechanism [15]. More recently, it was reported that calceolarioside A induced moderate antibacterial activity against *Bacillus cereus NRRLB 3711* [16], antifungal activity against *Malassenzia* spp. [17], and collagenase inhibitory activity [18] and putative anti-viral activity against COVID-19 from docking and molecular dynamics simulation data [19].

Other literature data on calceolarioside A may help us to provide a hypothesis on the mechanism by which this compound induces the antinociceptive and anti-inflammatory effects observed in our experiments. In a study examining the protective effect on adriamycin-induced cardiomyocyte toxicity, calceolarioside A significantly inhibited the adriamycin-induced cell death and caspase-3 activation, decreased the level of intracellular reactive oxygen species, and was more effective than those observed with the other antioxidants, including probucol, ascorbic acid, and alpha-tocopherol [20]. Another study later confirmed the antioxidant properties of calceolarioside A. As a part of research for antioxidative constituents from the genus *Buddleja*, Ahmad et al. [21] investigated the antioxidative activities of calceolarioside A isolated from *Buddleja davidii*. In fresh prepared rat kidney homogenates, calceolarioside A displayed strong scavenging potential for HO^•^, total ROS, and scavenging of ONOO– [21]. Production of ROS and reactive nitrogen species (RNS) is central to the progression of many inflammatory diseases, and ROS act as both a signaling molecule and a mediator of inflammation [22]. ROS produced in oxidative metabolism and some natural or artificial chemicals have been reported to initiate the inflammatory process, resulting in the synthesis and secretion of proinflammatory cytokines [22]. The activation of nuclear factor-kappa B/active protein-1 (NF-*κ*B/AP-1) and production of pro-inflammatory cytokines, mainly IL-1β, IL-12, IL-6, and TNF-α, have been, for instance, documented to play a critical role in the inflammatory and pain process, resulting in several chronic diseases [23]. In a recent study, and in line with what we observed in the experiments conducted on THP-1 cells, calceolarioside A was not cytotoxic to either mouse spleen cells or U266 cells, and strongly inhibited IgE production in U266 cells and IL-2 production in mouse spleen cells, in a dose-dependent manner [24].

Thus, the antioxidant effects of calceolarioside A could explain the antinociceptive and anti-inflammatory effects observed in our in vivo and in vitro experiments. However, other hypotheses can also be formulated. In 1988, Zhou and collaborators [25] reported the isolation of several phenylpropanoid glycosides from *Digitalis purpurea* and *Penstemon linarioides*, including calceolarioside A. These compounds were tested for their inhibitory activity against PKCα, and calceolarioside A was found to be the most active in inhibiting PKCα with IC_50_ values of 0.6 mM. PKCα has been implicated in various cellular functions, including proliferation, apoptosis, differentiation, motility, pain, and inflammation. The roles of central PKC in various pain states have intensively been investigated during the past decade. PKCα is an essential regulator of ion channel and membrane excitability in DRG neurons [26], and it has been reported that phosphorylated PKC increases in DRGs of MIA-induced rat joint pain [27]. Interestingly, the intraplantar injection of chelerythrine chloride, a PKC inhibitor, dose-dependently inhibited bee venom-induced nociception and inflammation in rats [28], and these findings resemble that which occurred in our in vivo experiments after calceolarioside A administration.

## 4. Materials and Methods

### 4.1. Drugs

Calceolarioside A (Figure 1) was obtained from *Calceolaria hypericina* Poepp. ex D. C., as previously reported [9]. The aerial parts of *Calceolaria hypericina* Poepp. ex D. C. were collected in Cuesta Zapata, V region, Chile, and identified at the Universidad Federico Santa Maria, Valparaiso, Chile where a specimen was deposited. In the general procedure, the aerial parts were extracted with EtOH at room temperature and, after evaporation, the residue of the extract separated by counter-current distribution with a biphasic solvent system composed of EtOAt:n-BuOH:H_2_0 in a suitable composition. The presence of calceolarioside A and other phenylpropanoid glycosides was determined by HPLC [9]. Calceolarioside A was stored at −20 °C until its use for in vivo and in vitro experiments. Carrageenan and zymosan A were purchased from Sigma-Aldrich (Milan, Italy). Dimethyl sulfoxide (DMSO) and formalin were purchased from Merck (Rome, Italy). Unless otherwise stated, all the other reagents were purchased from Carlo Erba (Milan, Italy).

### 4.2. Animals and Experimental Protocols

Male CD-1 mice (Harlan, Italy) of 3–4 weeks (25 g) were used for all the experiments. Mice were housed in colony cages, under standard conditions of light, temperature, and relative humidity for at least 1 week before starting experimental sessions. All experiments were performed according to Legislative Decree 27/92 and approved by the local ethics committee (Approval number 198/2013-B). Animal studies were performed in accordance to the ARRIVE guidelines [29].

### 4.3. Hot Plate Test

The hot-plate test was performed as described earlier [30]. A transparent plastic cylinder (14 cm diameter, 31 cm height) was used to confine the mouse on the heated (55 ± 0.5 °C) surface of the plate. The baseline latency was calculated as a mean of three readings recorded before testing at intervals of 15 min. The animals were placed on the hot plate 15 min after the i.c.v. injection of the vehicle (DMSO 1% solution in saline) or calceolarioside A (1, 5, and 10 μg), and the latencies to paw licking, rearing, or jumping were measured 15, 30, 45, and 60 min after administration. A cut-off time of 60 s was used to avoid tissue injury. The injection volume was 5 μL/mouse.

### 4.4. Tail Flick Test

The tail flick latency was obtained using a commercial unit (Ugo Basile, Gemonio, Italy), consisting of an infrared radiant light source (100 W, 15 V bulb) focused onto a photocell utilizing an aluminum parabolic mirror [31,32]. During the trials, the mice were gently hand-restrained using leather gloves. Radiant heat was focused 3–4 cm from the tip of the tail, and the latency (s) of the tail withdrawal to the thermal stimulus was recorded. The measurement was interrupted if the latency exceeded the cut-off time (15 s). The baseline latency was calculated as the mean of three readings recorded before testing at intervals of 15 min, and the time course of latency was determined at 15, 30, 45, and 60 min after i.c.v. injection of vehicle (DMSO 1% solution in saline) or calceolarioside A (1, 5, and 10 μg/5 μL). The injection volume was 5 μL/mouse.

### 4.5. Formalin Test

The procedure used has been previously described [33,34]. Subcutaneous (s.c.) injection of a dilute solution of formalin (1%, 20 μL/paw) into the mice hind paw evokes nociceptive behavioral responses, such as licking or biting the injected paw, which are considered indices of pain. The nociceptive response shows a biphasic trend, consisting of an early phase occurring from 0 to 10 min after the formalin injection, due to the direct stimulation of peripheral nociceptors, followed by a late prolonged phase occurring from 10 to 40 min, which reflects the response to inflammatory pain. During the test, the mouse was placed in a plexiglass observation cage (30 cm × 14 cm × 12 cm), 1 h before the formalin administration to allow it to acclimatize to its surroundings. Immediately after the formalin injection, the mouse was returned to the plexiglass observation cage, and nociceptive behavior was continuously measured using a stopwatch for 5 min intervals for a total testing time of 40 min. The total time(s) that the animal spent licking or biting its paw during the formalin-induced early and late phase of nociception was recorded. Calceolarioside A was dissolved in DMSO:saline (ratio 1:3 *v*/*v*) and then administered s.c. into the mice hind paw at a dose of 10, 50, and 100 μg in a volume of 20 μL, 30 min before the formalin (1%, 20 μL/paw).

### 4.6. Carrageenan-Induced Thermal Hyperalgesia

The plantar test (Ugo Basile, Italy) was used to measure the sensitivity to a noxious heat stimulus to assess thermal hyperalgesia after carrageenan administration [34,35]. A constant radiant heat source was directed on a mouse footpad until its withdrawal, foot drumming, or licking. A timer started automatically when the heat source was activated and a photocell stopped the timer when the mouse withdrew its hind paw. Animals were acclimatized to their environment for 1 h before the measurements of paw withdrawal latency (PWL), when exploratory behavior had ceased. The heat intensity was adjusted to obtain a baseline between 10 and 15 s, and a 30 s cut-off was used to avoid tissue damage. A total of 3 readings were taken from each paw and averaged. Animals were first tested to determine their baseline PWL to respond; 2 h later, each animal received an s.c. injection of 20 μL of 1% carrageenan into the dorsal surface of the right hind paw. The PWL (s) of each animal to the plantar test was determined again at 1, 2, 3, 4, and 24 h after the carrageenan injection. Mice received an s.c. injection of calceolarioside A (10, 50, or 100 μg/20 μL paw) into the dorsal surface of the right hind paw, 2.5 h after carrageenan. Calceolarioside A was dissolved in DMSO:saline (ratio 1:3 *v*/*v*).

### 4.7. Zymosan-Induced Paw Edema

Mice received an s.c. administration (20 μL/paw) of zymosan A (2.5% *w*/*v* in saline) into the dorsal surface of the right hind paw [36]. Paw volume was measured 3 times before the injections and at 1, 2, 3, 4, and 24 h thereafter using a hydroplethysmometer apparatus (Ugo Basile, Italy). The increase in paw volume was then evaluated as the percentage difference between the paw volume at each time point and the basal paw volume. Calceolarioside A dissolved in DMSO:saline (ratio 1:3 *v*/*v*) at increasing doses (1, 10, or 100 μg/20 μL paw) was administered s.c. into the dorsal surface of the right hind paw 30 min before zymosan.

### 4.8. Assay of Calceolarioside Anti-Inflammatory Activity on LPS-Stimulated Macrophage

Human peripheral blood monocytic cell line THP-1 was purchased from the American Type Culture Collection (Bethesda, MD, USA). Cells were maintained in RPMI-1640 medium supplemented with 2 mM L-glutamine and 100 U mL^−1^ of streptomycin-penicillin and 10% heat-inactivated fetal bovine serum (Sigma Aldrich, St. Louis, MO, USA) at 37 °C with 5% CO_2_. THP-1 cells were plated in 6-well culture plates at 1×10^6^ cells/well and were differentiated to macrophages using 100 ng mL^−1^ of phorbol-12-myristate-13-acetate (PMA, St. Louis, MO, USA) for 24 h with serum-free RPMI-1640 at 37 °C. After 72 h, the cells were treated with LPS at a final concentration of 0.1 μg mL^−1^ to stimulate cytokine production and with calceolarioside A at 10, 25, 50, and 100 μg mL^−1^. After 24 h of incubation, the supernatant was removed and centrifuged to remove any cell residues. The IL-6, IL-1β, and TNFα release was quantified using an ELISA, according to the manufacturer’s protocol (R&D Systems, Minneapolis, MN, USA).

In parallel, the effects of calceolarioside A on the viability of the LPS-stimulated macrophages were assessed using the 3-(4,5-dimethylthiazol-2-yl)-2,5-diphenyltetrazolium bromide (MTT) assay. After 24 h of exposure to calceolarioside A 1, 5, 10, 25, 50, 100, 250, and 500 μg mL^−1^, 20 μL of MTT (1 mg mL^−1^ in PBS) was added to each well and incubated continuously for 4 h under normal culture conditions. The cells were then treated with 100 μL of DMSO. The absorbance was measured at 570 nm using a microplate reader (Thermo MK3, Winosky, VT, USA). Data were expressed as a percentage of the value obtained for the solvent control (0.1% DMSO), which was set to 100%.

To reduce any variation from differences in cell density, the ELISA results were normalized to the MTT values. The concentration of cytokines of the positive control (cells only treated with LPS) was defined as 100%. All results from the tested calceolarioside A were then calculated as a percentage of the positive control [37].

## 5. Conclusions

In this study, we demonstrated for the first time the antinociceptive and anti-inflammatory effects of calceolarioside A. Calceolarioside A induced stimulus-dependent antinociceptive effects, as it reduced nociception induced by inflammatory stimuli but not by thermal nociceptive stimuli. This effect could be secondary to its anti-inflammatory properties, demonstrated in vivo in an edema model and in vitro in human macrophages. These data confirm the analgesic and anti-inflammatory properties of PPGs and indicate that this class of natural compound might represent a new therapeutic strategy for fighting pain-associated inflammatory diseases.

## Figures and Tables

**Figure 1 molecules-27-02183-f001:**
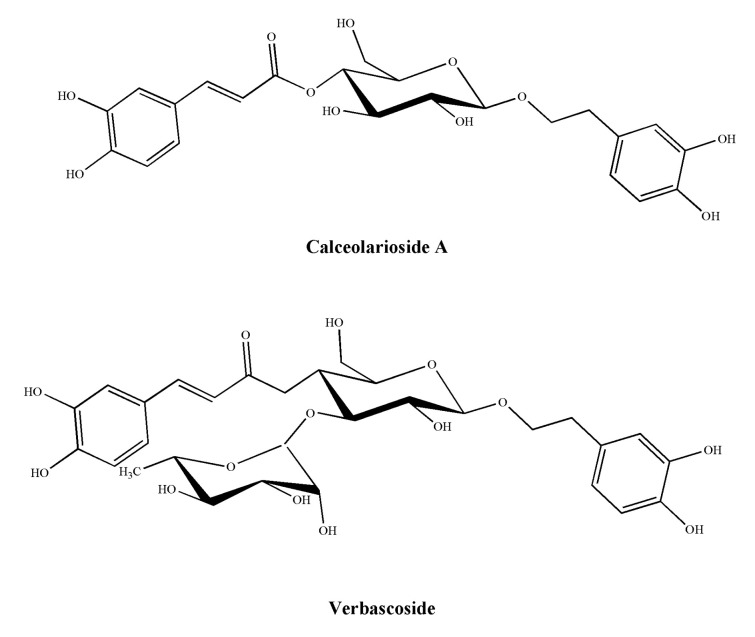
Structures of calceolarioside A and verbascoside.

**Figure 2 molecules-27-02183-f002:**
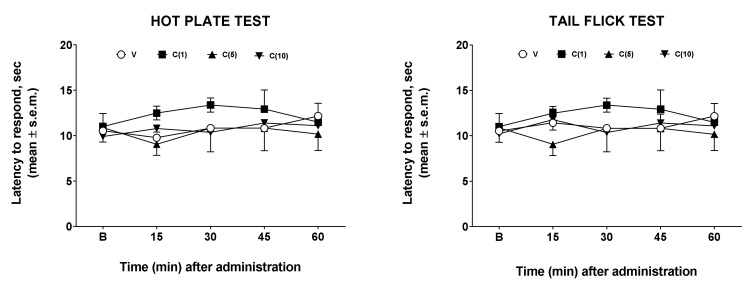
Effects induced by calceolarioside A administered i.c.v. at doses of 1, 5, and 10 μg (C(1–10)) in the hot plate (**left panel**) and tail flick test (**right panel**). Calceolarioside A did not change the response to thermal nociceptive stimuli. Statistical analysis was performed by using two-way ANOVA followed by Dunnett’s multiple comparisons test. N = 7.

**Figure 3 molecules-27-02183-f003:**
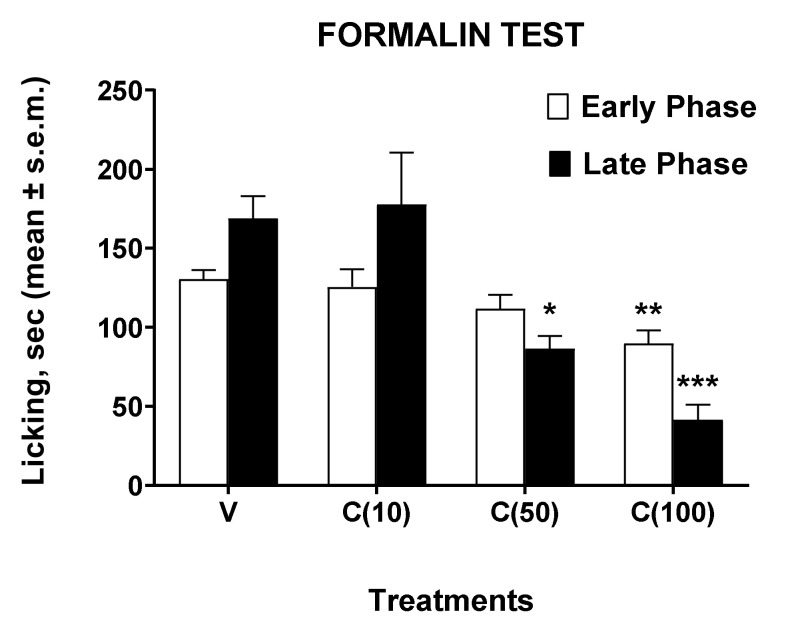
Effects induced by calceolarioside A administered s.c. into the mice hind paw at doses of 10, 50, and 100 μg (C(10–100)), 30 min before the formalin (1%, 20 μL/paw). Statistical analysis of the early and late phase was performed by using one-way ANOVA followed by Dunnett’s multiple comparisons test. * *p* < 0.05, ** *p* < 0.01, and *** *p* < 0.001 vs. V (vehicle-treated animals). N = 7.

**Figure 4 molecules-27-02183-f004:**
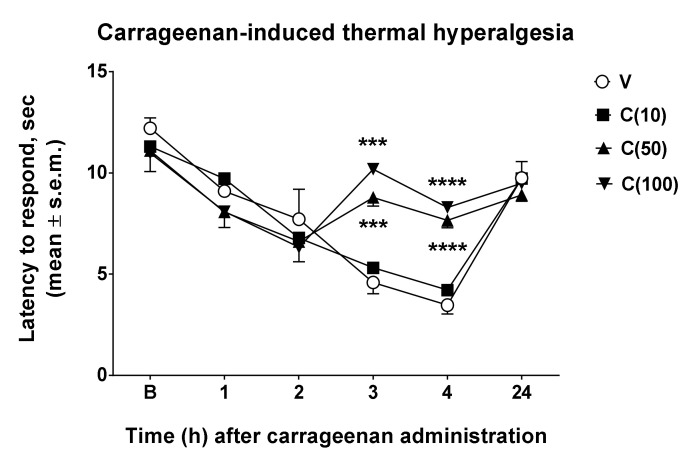
Effects induced by calceolarioside A on carrageenan-induced thermal hyperalgesia. Calceolarioside A was administered s.c. into the mice hind paw at doses of 10, 50, and 100 μg (C(10–100)), 2.5 h after carrageenan (20 μL of 1% carrageenan) administration in the same paw. Statistical analysis was performed by using two-way ANOVA followed by Dunnett’s multiple comparisons test. *** *p* < 0.001 and **** *p* < 0.0001 vs. V (vehicle-treated animals). N = 7.

**Figure 5 molecules-27-02183-f005:**
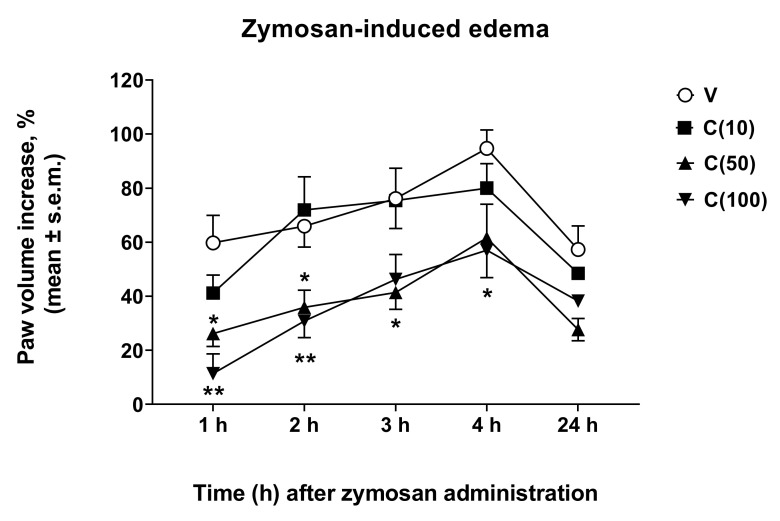
Effects induced by calceolarioside A administered s.c. into the mice hind paw at doses of 10, 50, and 100 μg (C(10–100)), 30 min before zymosan (2.5% *w/v* in saline, 20 μL/paw) administration in the same paw. Statistical analysis was performed by using two-way ANOVA followed by Dunnett’s multiple comparisons test. * *p* < 0.05 and ** *p* < 0.01 vs. V (vehicle-treated animals). N = 7.

**Figure 6 molecules-27-02183-f006:**
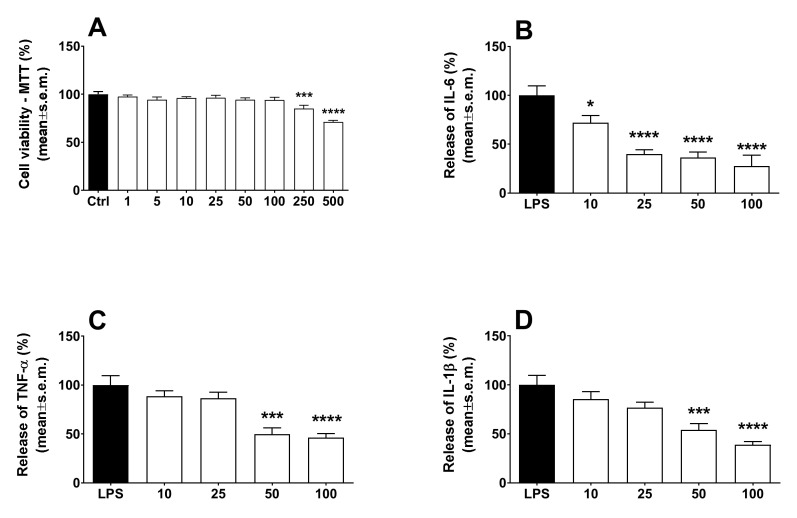
Effects induced by calceolarioside A on cell viability (**A**) and LPS-induced IL-6 (**B**), TNF-α (**C**), and IL-1β (**D**) release in macrophages. Macrophages were exposed for 24 h to 10, 25, 50, and 100 μg mL^−1^ of calceolarioside A. Results were obtained from three separate experiments performed in duplicate and reported as mean ± SEM. Statistical analysis was performed by one-way ANOVA followed by Dunnett’s multiple comparisons test. * *p* < 0.05, *** *p* < 0.001, and **** *p* < 0.0001 vs. LPS.

## Data Availability

Data relating to this research are available upon request.

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
