# Peer review of "Calceolarioside A, a Phenylpropanoid Glycoside from Calceolaria spp., Displays Antinociceptive and Anti-Inflammatory Properties"

_molecules, 2022, doi:10.3390/molecules27072183_

Round 1

Reviewer 1 Report

Dear authors,

The manuscript entitled "Calceolarioside A, a phenylpropanoid glycoside from Calceolaria spp displays antinociceptive and anti-inflammatory properties” reports the antinociceptive and anti-inflammatory effects of Calceolarioside A, a phenylpropanoid glycoside previously isolated from various Calceolaria species. It presents scientific relevance for the area of Medicine, Biology, Chemistry and natural products area. After consulting www.sciencedirect.com and https://pubmed.ncbi.nlm.nih.gov/, all authors have publications related to the subject of the manuscript. The language (English) are satisfactory (I suggest the final revision)! However, you need to change some details/informations in the abstract, Introduction, Methods, results, discussion and conclusions. 

Abstract: Adequate, but I suggest rewrite and entering some methodological information and the most expressive numerical results.

- Keywords: the words “Natural compounds” and “pain” do not appear in the abstract or title! I suggest reviewing!

* Introduction section: It is well written, but I suggest highlighting the "innovative" proposal of the study, as well as the advantages / disadvantages, at the end of the introduction.

* Results section

Wouldn't it be more interesting to combine the "results” with the "discussion" to better describe the findings and compare them with other works published in the literature? I suggest expanding the discussions!

- Page 4, lines 106, 110, 126...: To replace “h” by "hours". To review throughout the manuscript, if necessary.

- The global results obtained are interesting! Congratulations on the study!

* Discussion section

Wouldn't it be more interesting to combine the "results” with the "discussion" to better describe the findings and compare them with other works published in the literature? I suggest expanding the discussions!

- I suggest expanding the discussions, based on the results obtained, comparing the findings with the literature, especially for “Calceolarioside A effects on LPS-induced cytokine release from human macrophage”.

- Page 7, lines 209 – 212: I suggest writing a separate section on conclusions! I suggest rewriting, improving the conclusions based on some comments. I suggest highlighting the advantages of the method and the study!

* Material and Methods section: The methods used are suitable for the manuscript.

- Page 7, in “4.1. Drugs” section: I suggest inserting more information about the collection of the samples (period, place, packaging, etc.), as well as storage until the moment of the analyzes.

- Page 8, line 236 and 248, in “4.3. Hot plate test” and “4.4. Tail Flick Test “ sections: What is the real concentration of this solution (10 µg/5µl) To replace “µg/µl” by " µg µL-1 ". To review throughout the manuscript, if necessary.

- Page 9, lines 293, 298, 298….: To replace “U/ml” and “μg/mL” by “U mL-1” and “μg mL-1”. To review throughout the manuscript, if necessary.

* Tables and Figures: Adequate! Please, to improve the quality of figures 1 and 6.

* References: Please, check if the references are in accordance with the journal's rules.

Author Response

We thank the reviewer for the helpful comments and suggestions that greatly improved our manuscript. The responses to reviewer's questions are detailed in italics below.

Dear authors,

The manuscript entitled "Calceolarioside A, a phenylpropanoid glycoside from Calceolaria spp displays antinociceptive and anti-inflammatory properties” reports the antinociceptive and anti-inflammatory effects of Calceolarioside A, a phenylpropanoid glycoside previously isolated from various Calceolaria species. It presents scientific relevance for the area of Medicine, Biology, Chemistry and natural products area. After consulting www.sciencedirect.com and https://pubmed.ncbi.nlm.nih.gov/, all authors have publications related to the subject of the manuscript. The language (English) are satisfactory (I suggest the final revision)! However, you need to change some details/informations in the abstract, Introduction, Methods, results, discussion and conclusions. 

Abstract: Adequate, but I suggest rewrite and entering some methodological information and the most expressive numerical results.

As you suggested, we rewrote the abstract entering some methodological information and some numerical results. Please, see the abstract in the revised version of the manuscript.

- Keywords: the words “Natural compounds” and “pain” do not appear in the abstract or title! I suggest reviewing!

We agree with you and we changed the key words in:  calceolarioside A; Nociception; Inflammation; Phenylpropanoid glycosides; Calceolaria.

* Introduction section: It is well written, but I suggest highlighting the "innovative" proposal of the study, as well as the advantages / disadvantages, at the end of the introduction.

We thank the reviewer for appreciating our introduction. As stated in the introduction, the innovative purpose of our study was verify the possible antinociceptive and anti-inflammatory effects of calceolarioside A, never carried out before. As you suggested, at the end of the introduction we have added a short sentence about the possible advantages of our study.

* Results section

Wouldn't it be more interesting to combine the "results” with the "discussion" to better describe the findings and compare them with other works published in the literature? I suggest expanding the discussions!

We understand that it might be interesting to combine the results with the discussion but we believe that if we leave the two sections separated, the reader can better appreciate the results, the first in the literature on the antinociceptive and anti-inflammatory effects of calciolarioside A. We have already discussed all the known data on calceolarioside A and useful to explain our results, even those not directly demonstrated by our experiments. Furthermore, another reviewer finds our discussion too long and speculative. 

- Page 4, lines 106, 110, 126...: To replace “h” by "hours". To review throughout the manuscript, if necessary.

As you suggested, we replaced “h” by “hours” throughout the manuscript.

- The global results obtained are interesting! Congratulations on the study!

We thank the reviewer for the kind comments on our study.

* Discussion section

Wouldn't it be more interesting to combine the "results” with the "discussion" to better describe the findings and compare them with other works published in the literature? I suggest expanding the discussions!

- I suggest expanding the discussions, based on the results obtained, comparing the findings with the literature, especially for “Calceolarioside A effects on LPS-induced cytokine release from human macrophage”.

We thank the reviewer for the comments and suggestions. As you suggested, the effects of calceolarioside A in relation to the release of inflammatory cytokines are reported in the discussion section of the revised version of the manuscript. Please, see page 7, lines 190-199.

- Page 7, lines 209 – 212: I suggest writing a separate section on conclusions! I suggest rewriting, improving the conclusions based on some comments. I suggest highlighting the advantages of the method and the study!

As you suggested, we add a new conclusion section in the revised version of the manuscript. Please, see page 7, lines 217-224.

* Material and Methods section: The methods used are suitable for the manuscript.

- Page 7, in “4.1. Drugs” section: I suggest inserting more information about the collection of the samples (period, place, packaging, etc.), as well as storage until the moment of the analyzes.

As you have suggested, the required information has been reported in the revised version of the manuscript.

- Page 8, line 236 and 248, in “4.3. Hot plate test” and “4.4. Tail Flick Test “ sections: What is the real concentration of this solution (10 µg/5µl) To replace “µg/µl” by " µg µL-1 ". To review throughout the manuscript, if necessary.

We agree with the reviewer that our way of reporting doses used in our experiments may be unclear. We have corrected this in the revised version of the manuscript and changed the way of expressing concentration, as you suggested.

- Page 9, lines 293, 298, 298….: To replace “U/ml” and “μg/mL” by “U mL-1” and “μg mL-1”. To review throughout the manuscript, if necessary.

As you suggested, in the revised version of the manuscript we changed the way of expressing concentration.

* Tables and Figures: Adequate! Please, to improve the quality of figures 1 and 6.

As you suggested, in the revised version of the manuscript we improved the quality of figures 1 and 6.

* References: Please, check if the references are in accordance with the journal's rules.

As you suggested, we checked the references in accord to the rules of the journal.

Reviewer 2 Report

The paper molecules-1634473 characterizes calceolarioside A, a plant-derived substance regarding antinociceptive and antiinflammatory activity. The methods are sound, and the results obtained with the different assays are congruent demonstrating the antihyperalgesic and antiinflammatory efficacy of the agent.

Questions:

1./ Why was calceolarioside A intracerebroventricularly applied in the hot plate and tail-flick tests?  It is especially strange because in the other test the drug was given topically, into the hind paw.

2./ When studying carrageenan-induced thermal hyperalgesia, the plantar test was used for assessing thermonociception. However, upon examination of the baseline heat sensitivity in control animals the hot plate and tail-flick tests were used. What explains this discrepancy?

3./ When the anti-edema effect of calceolarioside A was investigated, why was not the carrageenan model used? The other way around, why was not thermonociception assessed in the zymosan model?

Comments, suggestions

In case of Fig. 4, no information is provided regarding the thermonociceptive test used.

The part of the discussion dealing with the role of reactive oxygen species is very long and speculative. As no result is provided regarding this issue, this part could be drastically shortened.

Calceolarioside A does not need to be written with capital initial.

The English of the abstract must be corrected.

Author Response

We thank the reviewer for the helpful comments and suggestions. The responses to reviewer's questions are detailed in italics below.

The paper molecules-1634473 characterizes calceolarioside A, a plant-derived substance regarding antinociceptive and antiinflammatory activity. The methods are sound, and the results obtained with the different assays are congruent demonstrating the antihyperalgesic and antiinflammatory efficacy of the agent.

Questions:

1./ Why was calceolarioside A intracerebroventricularly applied in the hot plate and tail-flick tests?  It is especially strange because in the other test the drug was given topically, into the hind paw.

The amount of calceolarioside A to be used for our experiments was low. Plants produce these compounds in minimal quantities compared to other metabolites. Topical or i.c.v. administration requires minimal amounts of substance in confront to systemic administration. Furthermore, the hot plate and the tail flick test are sensitive to compounds with central activity even if administered systemically, and i.c.v. route appeared the right choice in our experimental conditions.

2./ When studying carrageenan-induced thermal hyperalgesia, the plantar test was used for assessing thermonociception. However, upon examination of the baseline heat sensitivity in control animals the hot plate and tail-flick tests were used. What explains this discrepancy?

The hot plate and tail flick test and the plantar test all measure animals’ response to thermal nociceptive stimuli. However, these tests are different in nature since responses in the hot plate test are considered supraspinal, the tail flick is a spinal reflex, but it is subject to supraspinal influences that can affect it and the plantar test differentiates the left and right hind paw responses in freely moving animals. Furthermore, the hot plate and the tail flick test are used for evaluate central acting drugs, whereas the plantar test is used for evaluate drugs that reduce hyperalgesia induced by local inflammation. For these reasons, we wanted to use these tests, also using different routes of administration to evaluate in the best possible way the antinociceptive effects of calceolarioside A.

3./ When the anti-edema effect of calceolarioside A was investigated, why was not the carrageenan model used? The other way around, why was not thermonociception assessed in the zymosan model?

We agree with the reviewer that we could have used carrageenan or zymosan to measure both edema and hyperalgesia. However, in our experience, carrageenan induces a less variable nociceptive threshold reduction than zymosan. For this reason, we decided to use carrageenan to evaluate the anti-hyperalgesic effects of calceolarioside. The experiments with zymosan, were conducted some time before those with carrageenan, and at that time the plantar test was not available in the laboratory.

Comments, suggestions

In case of Fig. 4, no information is provided regarding the thermonociceptive test used.

As you suggested, in the revised version of the manuscript we changed figure 4 and its caption, inserting information on the thermonociceptive test used.

The part of the discussion dealing with the role of reactive oxygen species is very long and speculative. As no result is provided regarding this issue, this part could be drastically shortened.

We agree with the reviewer that our discussion is essentially speculative. But the data in the literature on calceolarioside A are very few, and we wanted to report and discuss all the few data useful to explain our results and the possible mechanism of action of calceolarioside A, including those relating to reactive oxygen species. However, as you suggested, we reduced discussion dealing with the role of reactive oxygen species. Please, see the revised version of the discussion. 

Calceolarioside A does not need to be written with capital initial.

As you suggested, in the revised version of the manuscript  we changed Caleolarioside in calceolarioside. We have also changed the other natural products mentioned in the text accordingly.

The English of the abstract must be corrected.

We apologize for the errors in writing the abstract, that were corrected in the revised version of the manuscript.